# A Task Scheduling Algorithm for Phased-Array Radar Based on Dynamic Three-Way Decision

**DOI:** 10.3390/s20010153

**Published:** 2019-12-25

**Authors:** Bo Li, Linyu Tian, Daqing Chen, Yue Han

**Affiliations:** 1School of Electronics and Information, Northwestern Polytechnical University, Xi’an 710129, China; tianly@mail.nwpu.edu.cn (L.T.); 17802933853@mail.nwpu.edu.cn (Y.H.); 2School of Engineering, London South Bank University, London SE1 0AA, UK; chend@lsbu.ac.uk

**Keywords:** comprehensive priority, phased-array radar, three-way decision, task scheduling

## Abstract

The time resource management of phased-array radars is the key to fulfilling their performance, such as how phased-array radar can efficiently and reasonably schedule tasks under limited resources. Therefore, this paper proposes a task scheduling algorithm for phased-array radar based on dynamic three-way decision. The algorithm introduces three-way decision into the scheduling algorithm and divides the target into three threat areas according to the threat degree (i.e., threat area, nonthreat area, and potential threat area). Different threat domains are assigned different weights and combine the working mode and the task deadline to carry out comprehensive priority planning, so that the radar can reasonably allocate time according to the difference of the target threat level and the threat area in the tracking stage. In addition, an improved adaptive threshold algorithm is proposed to obtain a dynamic three-way decision to achieve the adaptation of the algorithm. A set of performance indicators have been defined to evaluate the algorithm. The relevant experiments have demonstrated that the proposed algorithm can effectively improve the processing capability of phased-array radars when dealing with high-threat targets.

## 1. Introduction

A phased-array radar has flexible, fast beam agility and beam adaptability. It can perform searching and tracking tasks simultaneously and can track multiple targets at the same time. These features have made phased-array radars more powerful compared with traditional radars with a mechanical beam-steering antenna. However, realizing the advantages of phased-array radars depends on an efficient strategy for radar time resources allocation, which involves assigning a specific task to each time division to achieve an optimal use of limited radar time resources. Therefore, it is very important to study how to achieve an optimized allocation of tasks with limited time resources so as to fulfill the advantages of the phased-array radar. 

In general, the phased-array radar task scheduling module can be divided into two parts: task priority planning and selection of scheduling strategy.

Traditional task priority planning algorithms assign priorities at different levels according to the type of task in advance, and the priorities are fixed during the scheduling process. As a result, those algorithms have a poor adaptability. With the introduction of a dynamic scheduling priority algorithm, such as the Earliest Deadline First algorithm (EDF) [1,2,3], radar scheduling performance has been improved. However, the problem with this algorithm is that the condition of determining the priority is single, and the deadline of a task is the only criterion for determining the priority of a task. Priority given by considering multiple factors is a comprehensive priority. Today, in task priority planning, multiple factors of a task are often considered collectively to get a comprehensive priority of a task. Typically there are two commonly used comprehensive priority algorithms: (1) the algorithm of High Priority and Earliest Deadline First (HiPrEDF) [4,5,6], which maps the deadline and working mode priority of a task to a unified level to calculate the comprehensive priority of a task; and (2) the algorithm of Threat, High Priority, and Earliest Deadline First (THiPrEDF) [7,8,9], which fully utilizes the target information detected by the radar and combines the target threat degree with the priority of the radar task to determine the task comprehensive priority. The HiPrEDF algorithm considers the deadlines and the priority of different task types to calculate the comprehensive priority, and then conducts scheduling. A simulation of the HiPrEDF algorithm was undertaken in [5] and it concluded that the scheduling performance of the algorithm was better than an algorithm that only considers a single factor. In [6], various factors, such as task requests and scheduling restrictions, were integrated into scheduling by using the idea of the HiPrEDF algorithm for reference and achieved good results. In [7,8,9], a target threat degree was introduced as a factor into the calculation process of comprehensive priority. According to a target’s attributes, a quantitative model of threat degree was given and introduced into the phased-array radar adaptive scheduling algorithm. The target threat degree was combined with the working mode and the task deadline to carry out comprehensive priority planning, so that the tracking task priority can change according to the target threat degree. 

The HiPrEDF algorithm does not consider factors about targets, such as the target threat. Although the THiPrEDF algorithm has introduced the information of target threats, the algorithm does not consider that targets with different threat levels have different importance in task scheduling. For a target with low threat, the priority may not be higher than the priority of the search or confirmation tasks; however, considering its target threat level, the comprehensive priority of the task could be higher than other search or confirmation tasks. Therefore, it will perform tracking tasks with low actual value of scheduling, rather than searching or confirming tasks with a higher value, resulting in wasted time resources. In this paper, a new comprehensive priority algorithm is provided based on the three-way decision theory [10] to deal with the target threat degree and to assign targets into a threat area: a potential threat area or a no-threat area. We have carried out different comprehensive priority planning activities according to different threat areas. We also propose an improved three-way decision threshold algorithm to obtain a dynamic three-way decision model to better deal with the target threat degree. The proposed comprehensive priority algorithm has also been applied to adaptive scheduling in a multitask environment and combined with the scheduling strategy based on time pointer analysis [11].

The remainder of this paper is organized as follows. Section 2 presents radar mission modeling and the existing comprehensive priority. Section 3 gives the dynamic model of three-way decision and our new improved comprehensive priority planning algorithm. Section 4 gives indicators for evaluating the performance of the task scheduling algorithm proposed in this paper. The analytical experiments and the relevant results obtained are given in Section 5, and finally, the concluding remarks are summarized in Section 6. 

## 2. Radar Task and Comprehensive Priority Planning

Before discussing the comprehensive priority planning, we first have a look at radar task models. The radar tasks can be categorized into four types: search, confirmation, tracking, and miss. The sequence of the tasks requested is generally: search-confirmation-tracking (-miss-tracking maintenance). Assuming that there are *n* radar tasks, the *j*th (j=1,2,⋯,n) radar task can be simply described as: (1)$$T_{j}=\{P_{j},t_{aj},t_{dwj},t_{wj},t_{dj},\Delta t_{j},Num_{j},I_{nj}\}$$
where $P_{j}$ is the task type, $t_{aj}$ is the task request time [12], the time by which a task requests the radar to schedule, $\Delta t_{j}$ is the sample interval between two tasks [12], $t_{dwj}$ is the task dwell time [13], and $t_{wj}$ is the time window [14,15,16]. It is the range of time during which the actual execution time of the radar task can move back and forth relative to the expected execution time. $t_{dj}$ is the task deadline, which means the time by which the task stops being scheduled, $Num_{j}$ is the number of times the task has been scheduled, and $I_{nj}$ is the information of targets detected by the radar. 

The request time for each task is:(2)$$t_{aj}=t_{e(j-1)}+\Delta t_{j}$$
where $t_{e(j-1)}$ is the moment at which the last task has been successfully executed.

The deadline of a task can be expressed as:(3)$$t_{dj}=t_{aj}+t_{wj}$$

The comprehensive priority is divided into two levels. The first level describes the priority of a target task with different threats of the same type. The first priority of the *j*th task is determined by working mode priority $p_{rk}$ and target threat $p_{tk}$:(4)$$p_{k1}=p_{rk}+\lambda_{tk}p_{tk}$$
where $\lambda_{tk}$ is the weight of a threat area.

The HiPrEDF does not consider the impact of the target threat in determining the comprehensive priority of tasks, and hence in the HiPrEDF, there is always $\lambda_{tk}=0$. In the THiPrEDF, the value $\lambda_{tk}$ is determined according to the type of a task. In general, the search and the confirmation tasks have no prior knowledge and do not involve the concept of target threat. The first-level priority of the search and the confirmation tasks are determined only by the priority of a working mode and there is $\lambda_{tk}=0$. For the miss and tracking tasks that involve target threat, there is always $\lambda_{tk}=1$. Therefore, the THiPrEDF calculates the comprehensive priority according to Equation (5):(5)$$p_{k1}=\left\{ \begin{array}{ll} p_{rk} & \mathrm{for}\;\mathrm{the}\;\mathrm{search}\;\mathrm{or}\;\mathrm{confirmation}\;\mathrm{task}\\ p_{rk}+p_{tk} & \mathrm{for}\;\mathrm{the}\;\mathrm{miss}\;\mathrm{or}\;\mathrm{tracking}\;\mathrm{task} \end{array}\right.$$

After the first-level priority has been set up, the second-level priority will be determined based on the results of the first-level priority and the task deadline. There are two main criteria for the second-level priority planning: (1) the earlier the deadline, the higher the priority of a task; and (2) the higher the importance of a task, the higher the priority. 

In the scheduling period, all tasks are sorted by the first-level priority from the smallest to the highest and the deadline in an inverse order. It is assumed that the position of the *i*th task in the first-level priority queue is n, and in the deadline queue is m; the larger the m or n, the higher the priority. The comprehensive priority function is determined as:(6)pric=km+(1−k)n
where *k* is a constant coefficient between 0 and 1, pric is the comprehensive priority, and the larger the pric, the higher the priority of a task and the sooner it will be scheduled by the radar.

## 3. Proposed Algorithm for Phased-Array Radar Task Scheduling

### 3.1. Dynamic Three-Way Decision Model

Suppose that U is a collection of events, x is an event object, and x∈U. Then, x can be expressed with either a membership description P or a nonmembership description N [17,18]. Based on the two types of descriptions P and N, the whole domain U can be divided into two subsets: a membership set and a nonmembership set, represented by L and $L^{R}$, respectively. We use μ(x) and ν(x) to describe a membership and a nonmembership [19] of object x.

If there is a pair of thresholds α and β, where 0≤β<α≤1, based on μ(x) the object x may be assigned into one of the three regions: the positive region (*POS*), boundary region (*BND*), and negative region (*NEG*) as
(7)$$\begin{array}{l} POS\left(X\right)=\left\{x\in U|\mu \left(x\right)\geq \alpha \right\}\\ BND\left(X\right)=\left\{x\in U|\alpha <\mu \left(x\right)<\beta \right\}\\ NEG\left(X\right)=\left\{x\in U|\mu \left(x\right)\leq \beta \right\} \end{array}$$

Decision makers generally make different decisions based on the region to which an object belongs. The decision rules are as follows:

If the object *x* belongs to *POS*, the decision maker accepts the object;

If the object *x* belongs to *NEG*, the decision maker rejects the object;

If the object *x* belongs to *BND*, the decision maker chooses to delay the decision.

According to Equation (7), the decision-making action is determined by the value of the membership function μ(x) of the object *x* and the thresholds α, β [19,20,21,22]. Therefore, the determination of the membership function μ(x) and the determination of the thresholds α, β are particularly critical. 

In this paper, the membership function $\mu (x_{i})$ represents the threat level of the *i*th target $x_{i}$. The threat assessment algorithm based on intuitionistic fuzzy set [23] is used to obtain the weight of target attributes, and then the TOPSIS (Technique for Order Preference by Similarity to an Ideal Solution) model [24] is used to assess target threats. For the details of the threat assessment algorithm, please refer to [25].

Furthermore, we propose an improved algorithm for thresholds in order to determine better thresholds α and β. In the process of three-way decision, we may assign an object into a wrong region different from the region that it actually should be. The risk of this misclassification can be expressed as
(8)$$R_{X}\left(\alpha ,\beta ,\gamma \right)=\sum_{\mu \left(x_{i}\right)\geq \alpha }(1-\mu \left(x_{i}\right))+\sum_{\mu \left(x_{j}\right)\leq \beta }\frac{1-\gamma }{\gamma }\mu \left(x_{j}\right)+\varepsilon \sum_{\beta <\mu \left(x_{k}\right)<\alpha }\left[\frac{\beta (\alpha -\gamma )}{\gamma (\alpha -\beta )}(1-\mu \left(x_{k}\right))+\frac{\left(1-\alpha )(\gamma -\beta \right)}{\gamma (\alpha -\beta )}\mu \left(x_{k}\right)\right]$$
where 0≤β<γ<α≤1, ε≥1 and ε is a penalty factor to avoid dividing too many objects into a boundary region. According to the principle of minimizing a risk, the process of solving a threshold is equivalent to the process of solving the minimum value of $R_{X}$.

When determining the thresholds, the search space is defined as the set of the values of μ(x) of all objects. The ideas of the algorithm proposed in the paper are as follows:Step 1. Firstly, an initial value of the threshold is set up arbitrarily satisfying α>γ>β. Calculate the sum of the total risk $R_{X}$ with the given initial threshold and record it as $Mini_{R}$.Step 2. Sort out all objects according to the value μ(x) in descending order. Step 3. The thresholds (α,β,γ) are replaced by the value μ(x) of the object. All thresholds that meet α>γ>β are selected in turn and are reassigned as $\left(\alpha^{\prime },\beta^{\prime },\gamma^{\prime }\right)$.Step 4. Recalculate the total risk loss $R_{X^{\prime }}^{\prime }$ of the sample set with the new thresholds. If condition $R_{X^{\prime }}^{\prime }<Mini_{R}$ is satisfied, the threshold (α,β,γ) is updated with $\left(\alpha^{\prime },\beta^{\prime },\gamma^{\prime }\right)$; otherwise, the thresholds remain unchanged. Step 5. Determine whether all possible thresholds satisfying α>γ>β have been traversed completely. If so, execute Step 6; otherwise, go to Step 3 until all combinations have been traversed.Step 6. The final thresholds (α,β,γ) are the required result.

The pseudocode of Algorithm 1 is as follows:
**Algorithm 1:** The improved threshold algorithm.**Input:** Initial thresholds (α0,β0,γ0), α0>γ0>β0, the value μ(x) of all objects**Output:** Final thresholds (α,β,γ)Sort μ(x)=(μ(x1),μ(x2),⋯,μ(xn−1),μ(xn)), where μ(x1)>μ(x2)>⋯>μ(xn−1)>μ(xn)α←α0β←β0γ←γ0MiniR←RX(α0,β0,γ0)**For**
i←1 to n−2
**do** **For**
j←i+1 to n−1
**do**  **For**
k←j+1 to n
**do**   RX′′←RX(μ(xi),μ(xk),μ(xj))   **If** (RX′′<MiniR) **then**    α←μ(xi), β←μ(xk), γ←μ(xj)    MiniR←RX′′   **End**  **End** **End****End****Return**
(α,β,γ)

Based on the set of the thresholds (α,β,γ) and the threat of the targets, a target can be classified into one of these three areas: the threat area, the nonthreat area, or the potential threat area. More specifically, these areas can be expressed as:when $\mu (x_{i})\geq \alpha$, the target is classified into the threat area, and the target threat level is high;when $\mu (x_{i})\leq \beta$, the target is classified into the nonthreat area, and the target threat level is low or even negligible;when $\beta <\mu (x_{i})<\alpha$, the target is classified into the potential threat area, and the target is likely to move to the threat area or the nonthreat area at the next moment, depending on further observation of the target. 

According to the threshold algorithm discussed above, the thresholds can dynamically change with time due to different target threat degrees obtained at different time points. The results of the three-way decision classification of targets may be different at different times. Therefore, the algorithm can achieve dynamic classification of three-way decision.

### 3.2. Phased-Array Radar Task Scheduling Algorithm Based on Three-Way Decision

Based on the existing comprehensive priority planning, considering the target threat degree with three-way decision classification results, a new comprehensive priority planning algorithm is proposed: Threat with Three-Way Decision HiPrEDF (TWD-HiPrEDF). 

In the TWD-HiPrEDF, the dynamic results of the three-way decision classification of the target threat discussed in Section 3.1 are utilized to determine the comprehensive priority of the task. In Equation (4), $\lambda_{tk}$ takes different values according to the different areas where the target is classified. It can be expressed by the following formula:(9)$$\lambda_{tk}=\left\{ \begin{array}{ll} \lambda_{\alpha } & \mathrm{the}\;\mathrm{threatened}\;\mathrm{area}\\ \lambda_{\beta } & \mathrm{the}\;\mathrm{non}-\mathrm{threat}\;\mathrm{area}\\ \lambda_{\gamma } & \mathrm{the}\;\mathrm{potential}\;\mathrm{threat}\;\mathrm{area} \end{array}\right.$$

The values of $\lambda_{\alpha }$, $\lambda_{\beta }$, and $\lambda_{\gamma }$ are determined based on actual conditions of radar tasks, and the values satisfy $\lambda_{\alpha }>\lambda_{\gamma }>\lambda_{\beta }$. After determining the comprehensive priority of the task according to (4), it is necessary to determine whether the task is executed and what the execution time is according to the scheduling strategy. In this paper, we arrange the task request according to the level of comprehensive priorities within a scheduling interval. The flowchart of the TWD-HiPrEDF algorithm is shown in Figure 1. 

The specific steps of the scheduling algorithm are as follows:Step 1. Initialize the parameters of the scheduling interval: the length of the queue for the task request $N_{r}$, the time pointer $t_{p}$, and the end time of the scheduling interval $t_{e}$, and set i=0; check if the scheduling has completed. If completed, go to Step 7; otherwise, go to Step 2.Step 2. Remove a task from the request queue whose deadline is less than $t_{p}$, and count the number of tasks removed and record it as M. Set i=i+M.Step 3. Remove a task in the request task queue whose earliest executable time is less than $t_{p}$ and calculate their comprehensive priority to sort them. Select the task with the highest comprehensive priority as T.Step 4. When the dwell time of the task T is less than the remaining time of the current scheduling, move it to the execution queue and update parameters $t_{p}=t_{p}+t_{dw}$, i=i+1; otherwise, move it to the delay queue and go to Step 6.Step 5. If $t_{p}\geq t_{e}$ or $i\geq N_{r}$, go to Step 6; otherwise, go to Step 2.Step 6. Traverse the remaining request tasks. If conditions of the delay task are met, move it to the delay task queue, and $t_{aj}=t_{e}$, $t_{wj}=t_{wj}-(t_{e}-t_{aj})$; else move it to the delete queue.Step 7. The scheduling ends. The execution queue, the delay queue, and the delete queue are obtained.

The pseudocode of Algorithm 2 is as follows:
**Algorithm 2:** The radar scheduling algorithm:**Input:** The length of the queue for task request Nr, the requested tasks T={T1,T2,⋯,Tj,⋯,TNr}, where Tj={taj,tdwj,tdj,pricj}, the time pointer tp and the end time of the scheduling interval te**Output:** The execution queue Eq, the delay queue DYq, and the delete queue DTq
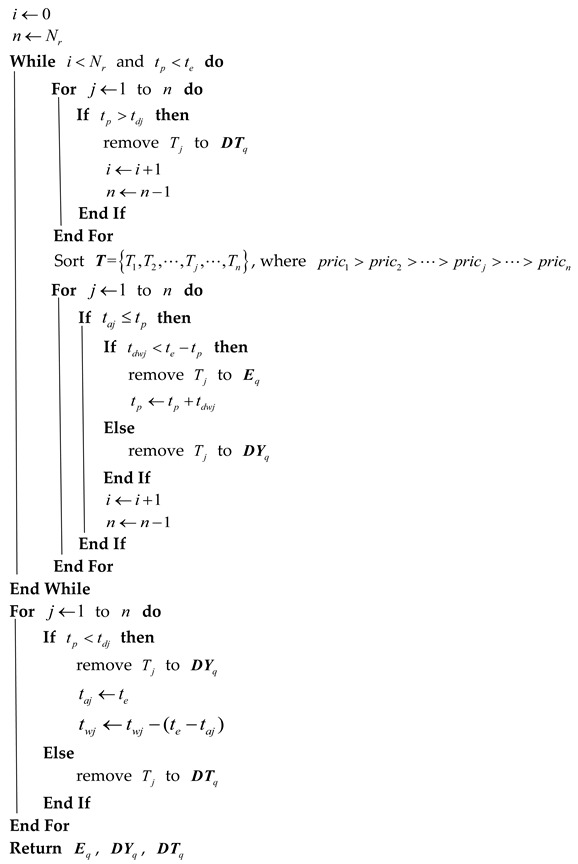


After the execution queue, the delay queue, and the delete queue are obtained by the scheduling algorithm, the radar tasks are sequentially executed. The state of a radar task is adjusted according to different situations when the task has been executed. A certain transformation is satisfied between the various states of the radar.

The tasks in the radar execution queue will be executed at the scheduling interval. After executed, the type of the task will change accordingly. The specific transformation of task types is: when the radar task is of the search type, if the target is detected, the task will become the confirmation type; if the target is not detected, the radar task will remain as the search type. When the radar task is of the confirmation type, if the target is detected, it will become the tracking type; if the target is not detected, it will become the search type. When the radar task is of the tracking type, if the target is detected, the tracking type is maintained; if the target is not detected, for the stable tracking task (the tracking number is greater than three times), it will become the miss type, and for the unsteady tracking task, it will not become the miss type. When the radar task is a miss type, if the target is detected, it becomes the tracking type; if the target is not detected, it becomes the search type. The specific transformation of task types is shown in Table 1.

The tasks in the radar delay queue are not executed during the scheduling interval and are delayed until the next scheduling interval. Therefore, the type of radar task does not change. The tasks in the radar delete queue are different. If the tracking type task is deleted, when the tracking time is less than or equal to two times, no miss processing will occur.

## 4. Performance Indicator

Phased-array radar resource management is mainly for the management of events and energy resources. Three main principles have been taken into account in evaluating a scheduling algorithm:

Priority principle: When the radar resources are insufficient and fail to satisfy all requests, the radar gives priority to tasks with higher priority. Therefore, if there is a time conflict between multiple radar tasks, the task with higher priority will be executed first. If a task with lower priority task cannot be scheduled within the scheduling interval, the task will be discarded or delayed. 

Time utilization principle: To improve the utilization of time resources, the idle time of a radar is required to approach 0, and therefore, more tasks could be completed in a scheduling interval. The principle is described as
(10)$$\left(SI-\sum_{i=1}^{N}\Delta t_{i}\right)\to 0$$
where SI is the length of a scheduled time, N is the number of tasks scheduled successfully, and $\Delta t_{i}$ is the dwell time of the *i*th task successfully scheduled.

Expectation time principle: This principle is mainly concerned with the actual execution time of a scheduled task, and it should be as close as possible to the expected execution time.
(11)$$\left|t_{si}-t_{ei}\right|\to 0$$
where $t_{si}$ is the actual execution time of the *i*th scheduled task, and $t_{ei}$ is the expected execution time of the *i*th scheduled task.

To evaluate the performance and defectiveness of a phased-array radar scheduling algorithm, a set of indicators should be adopted. In relation to these principles, there are several indicators that can be used to evaluate the performance and the effectiveness of a phased-array data scheduling algorithm as discussed below:(1)Total successful scheduling rate (SSR): It describes the degree of compliance of the scheduling algorithm with respect to the time utilization principle and the task scheduling.
(12)$$SSR=N_{suc}/N_{total}$$
where $N_{total}$ is the total number of tasks requesting scheduling, and $N_{suc}$ is the total number of tasks that are successfully scheduled.(2)Missed deadline rate of tasks ($MDR_{i}$): It relates to the degree of compliance of the scheduling algorithm with respect to the priority criteria and the scheduling situation of each task.
(13)$$MDR_{i}=N_{miss}/N_{totali}$$
where *i* has a value range of i=1,2,3,4, representing four types of radar missions, $N_{missi}$ is the number of *i*th radar tasks unsuccessfully scheduled by deadline, and $N_{totali}$ is the number of *i*th radar tasks requesting scheduling.(3)Threat rate of the execution (*TRE*): It describes the ratio of the sum of threats of the successfully scheduled tasks to the sum of threats of the total requested tasks. The scheduling performance of the scheduling algorithm for important tasks can be measured as
(14)$$TRE=\sum_{i=1}^{N_{suc}}P_{si}/\sum_{i=1}^{N_{sum}}P_{ti}$$
where $P_{si}$ is the threat degree of the *i*th scheduled successful task, and $P_{ti}$ is the threat degree of the *i*th scheduled task.

## 5. Results

### 5.1. Settings of Simulation Parameters

In the simulation experiments, a phased-array radar under consideration has four types of tasks: search, confirmation, tracking, and miss. The task parameters [8] are shown in Table 2.

During the simulation, a certain number of targets have been generated in the radar search sector and move to the radar at a uniform speed. When a target enters a radar’s detection range, it is searched by the radar with a certain probability. The radar uses the search and tracking method to detect targets and changes the state in real time according to detection results. 

The task scheduling interval was set to 50 ms, and the simulation duration had 500 scheduling intervals. The total number of the targets in the simulation was initialized as 10 and increased by 10 up to 80, and 100 simulations were performed for each additional 10 targets. The results were averaged.

The two algorithms, HiPrEDF and THiPrEDF, and the proposed TWD-HiPrEDF algorithm were simulated in the experiments. Among them, k=0.8 and the values of $\lambda_{\alpha }$, $\lambda_{\beta }$, and $\lambda_{\gamma }$ in the TWD-HiPrEDF algorithm were 1.5, 0.5, and 1.2, respectively. The performances of the three algorithms were measured according to performance evaluation indicators.

### 5.2. Discussion

According to the simulation parameters, the radar scheduling with the three algorithms was simulated separately. The obtained performance parameters are shown in Table 3, Table 4 and Table 5.

It can be seen from Table 3, Table 4 and Table 5 that the utilization rates of time for the three algorithms are not much different, and as the number of targets increases, the time resources can be fully utilized.

Figure 2 shows a comparison of the scheduling success rate of the three scheduling algorithms. It can be seen from the figure that when the target number was less than 40, the scheduling success rate of the three scheduling algorithms could reach 100%. When the target number reached 40, the scheduling success rate started to decline. When the number of targets reached 90, the scheduling success rate was less than 50%. As the number of targets increased, the downward trends of scheduling success rates of the three algorithms were similar. The scheduling success rate of the TWD-HiPrEDF was higher than the THiPrEDF, both of which were higher than the HiPrEDF. In conclusion, the proposed TWD-HiPrEDF algorithm has a higher scheduling success rate when the number of targets is greater than 60. Compared with the other two algorithms, the algorithm proposed can schedule more tasks without reducing the time utilization rate.

Figure 3 provides a comparison of the threat rate of the execution for three scheduling algorithms. It can be seen from the figure that when the target number was less than 40, the threat rate of the execution under the three scheduling algorithms could reach 100%. When the target number reached 40, the threat rate began to decrease. When the target number was between 40 and 90, the threat rates of executions of the TWD-HiPrEDF and the THiPrEDF were significantly higher than that of the HiPrEDF, indicating that the threat degree had been introduced into the scheduling algorithm, which significantly improved the scheduling of tasks with a higher threat level. At the same time, it has demonstrated that the TWD-HiPrEDF achieved a higher threat rate of the execution than the THiPrEDF when the target number was between 40 and 90. The results show that algorithms that introduce the target threat in the process of determining the comprehensive priority improve the task execution ability of the target with higher threat, that is, they can schedule higher priority and urgent tasks. The TWD-HiPrEDF scheduling algorithm further divides target threats into three areas: threat, potential threat, and no threat. In the process of introducing the threat degree into the comprehensive priority planning, different weights were introduced for the threat degree according to the different levels of target threats, and tasks with higher priority could be scheduled more effectively.

Figure 4 shows a comparison of missed deadline rates of tracking by the three scheduling algorithms. As shown in this figure, when the target number was less than 40, the missed deadline rate of tracking for the three scheduling algorithms was almost 0. When the target number was greater than 40, the missed deadline rate of tracking gradually began to increase. When the target number was greater than 60, the missed deadline rate of tracking had a significant increase. The missed deadline rate of tracking for the proposed TWD-HiPrEDF was significantly lower than that of the THiPrEDF, both of which were lower than the missed deadline rate of tracking for the HiPrEDF. The results show that the introduction of target threat degree and the three-way decision improves the tracking ability of the radar, enabling the radar to track the target more stably.

A further analysis on the other set of data was carried out in order to prove that the radar scheduling algorithm using the comprehensive priority planning with dynamic three-way decision could better improve the tracking performance of targets with high threat level. When the number of targets was 80, the number of targets in the tracking state in each scheduling interval was counted, and the number of targets in the 500 scheduling intervals was averaged to obtain the average number of targets in the tracking state within one scheduling interval and measure the performance of the scheduling algorithm for tracking tasks. At the same time, the average number of targets in the threat area, the potential threat area, and the nonthreat area was counted separately, which was used to measure the performance of the scheduling algorithm for tracking tasks of targets in different threat areas. 

All the simulation data was recorded in Table 6 and Figure 5, and the proportions of targets in different threat areas were recorded in Table 7.

As shown in Figure 5, compared with the traditional scheduling algorithm HiPrEDF, both the TWD-HiPrEDF and the THiPrEDF could track more targets in the threat area and the potential threat area, and fewer targets in the nonthreat area. At the same time, Table 7 shows that compared with the HiPrEDF, the TWD-HiPrEDF and the THiPrEDF had higher proportion of targets in the threat area and the potential threat area, and lower proportion of targets in the nonthreat area. In summary, the introduction of the target threat as a factor into the determination of comprehensive priority could improve the tracking performance for valuable targets and reduce the tracking performance of targets with no threat or low threat.

Further analysis of Figure 5 and Table 6 shows that the TWD-HiPrEDF algorithm could track more targets than the THiPrEDF in the three areas. At the same time, it has been found that compared with the HiPrEDF, the total number of targets under the THiPrEDF was reduced, and the total number of targets under the TWD-HiPrEDF was higher, indicating that the TWD-HiPrEDF can compensate for the shortage of reducing the total number of tracking targets in the THiPrEDF. In addition, compared with the other two algorithms, the TWD-HiPrEDF algorithm tracks the most targets in the threatened area. In summary, the proposed scheduling algorithm combining the threat assessment results with three-way decision can further deal with more targets with high threat and improve the tracking performance for targets with high threat while ensuring that the total number of tracking targets is not reduced.

## 6. Conclusions

The efficient phased-array radar scheduling algorithm is the key to improving the scheduling performance of phased-array radar. This paper introduces a dynamic three-way decision to deal with the target threat degree and proposes a new comprehensive priority algorithm applied to the scheduling strategy to obtain a new phased-array radar task scheduling algorithm. Compared with the existing two algorithms, the proposed algorithm can reduce the missed deadline rates of the tracking task and can improve the scheduling success rate of tasks and the threat rate of the execution. The introduction of the three-way decision enables the radar to schedule more important tasks more effectively and it can increase the proportion of targets with high threat and improve the tracking performance of the radar for targets with high threat while ensuring that the number of tracking of the overall target is not reduced. The simulation results have shown that the proposed algorithm can effectively improve the performance of phased-array radar task scheduling while ensuring target tracking. It provides a reference for the performance optimization research of phased-array radar.

## Figures and Tables

**Figure 1 sensors-20-00153-f001:**
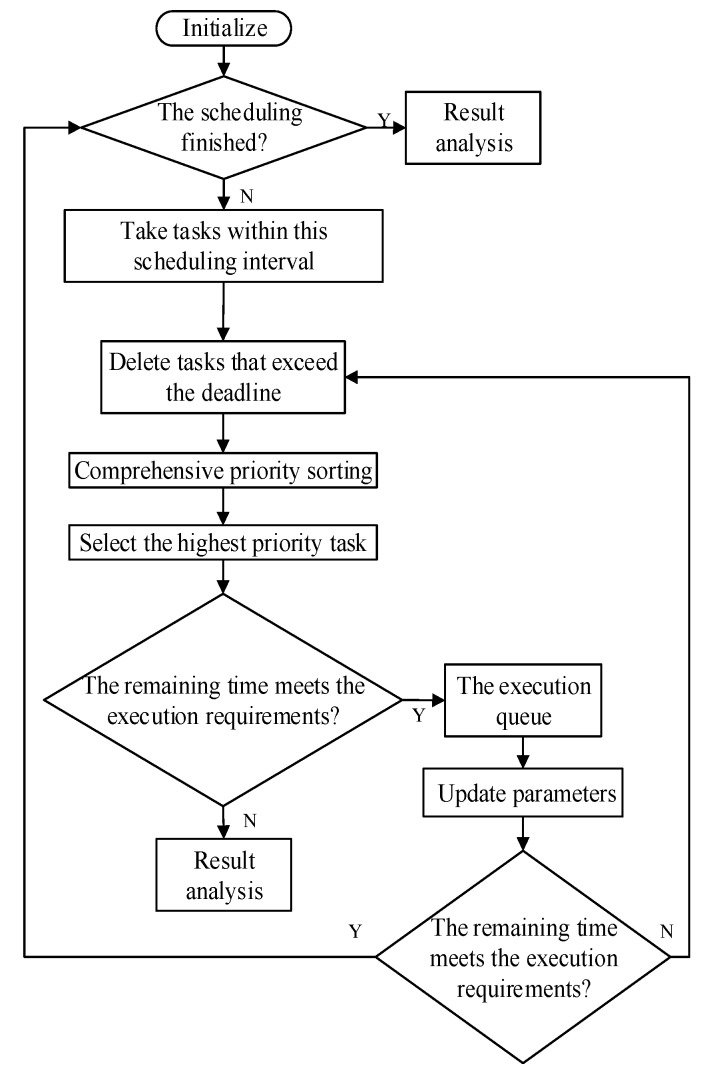
The flow chart of the proposed radar scheduling algorithm.

**Figure 2 sensors-20-00153-f002:**
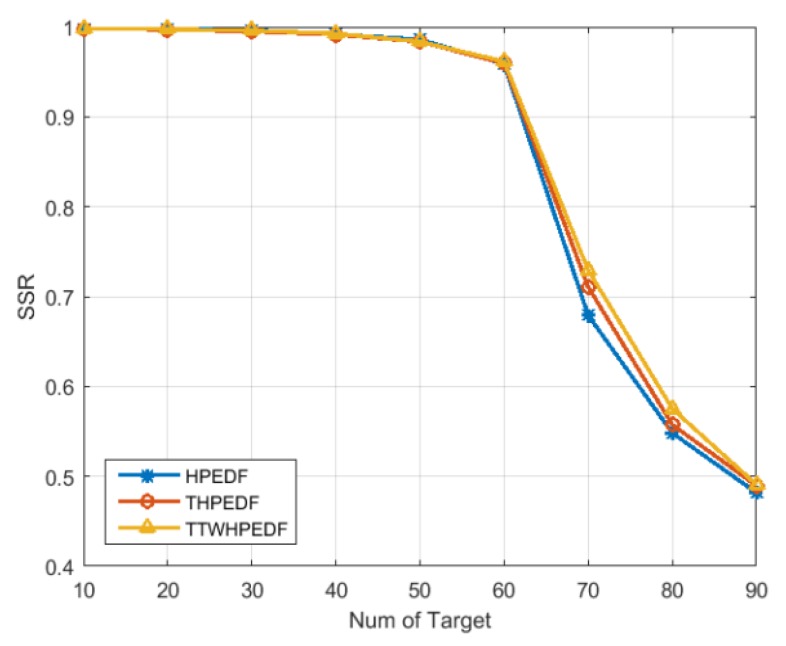
Comparison of scheduling success rates (*SSR*) of the three algorithms.

**Figure 3 sensors-20-00153-f003:**
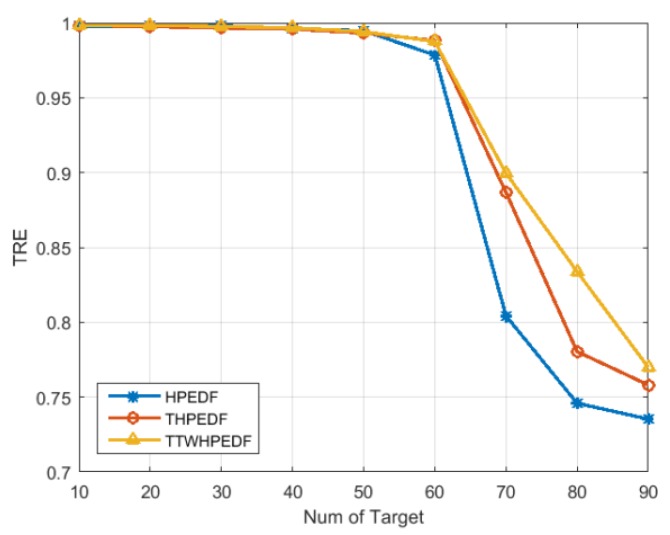
Comparison of the threat rate of the execution (*TRE*) for three scheduling algorithms.

**Figure 4 sensors-20-00153-f004:**
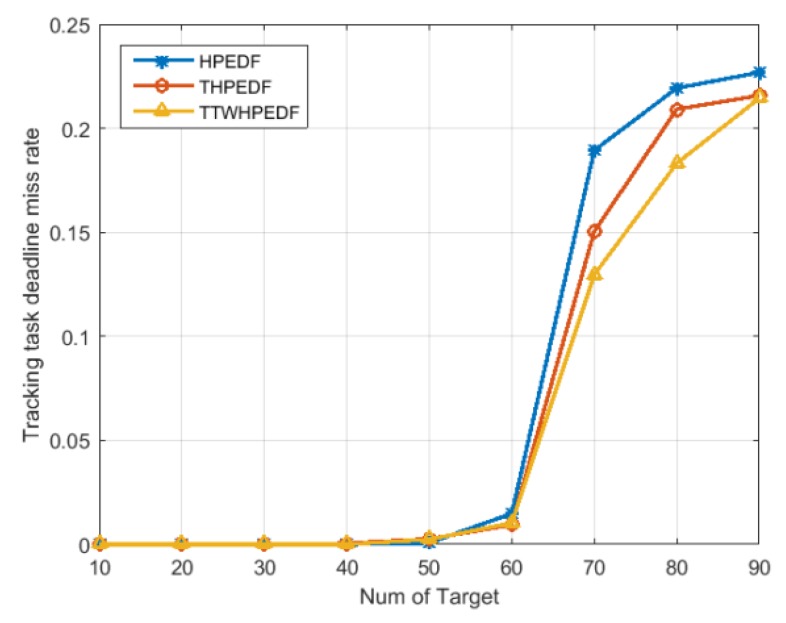
Comparison of missed deadline rates of tracking ($MDR_{2}$).

**Figure 5 sensors-20-00153-f005:**
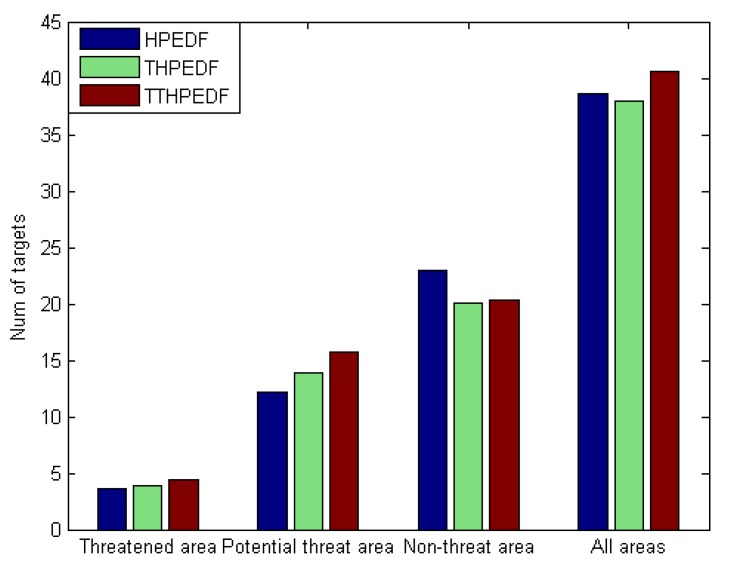
Number of targets in the tracking state under the three scheduling algorithms.

**Table 1 sensors-20-00153-t001:** The specific transformation of task types.

Task Type	Whether the Target Is Detected	Transformed Task Type
Search	Yes	Confirmation
	No	Search
Confirmation	Yes	Tracking
	No	Search
Tracking	Yes	Tracking
	No (the stable tracking)	Miss
	No (the unsteady tracking)	Tracking
Miss	Yes	Tracking
	No	Search

**Table 2 sensors-20-00153-t002:** Radar task parameters.

Task	The Working Mode	Dwell Time (ms)	Time Window (ms)	Sampling Interval (ms)
Confirmation	4	6	20	100
Miss	3	6	30	200
Tracking	2	2	50	200
Search	1	6	100	20

**Table 3 sensors-20-00153-t003:** Performance parameters of HiPrEDF.

Parameters/the Number of Targets	Total Successful Scheduling Rate (*SSR*)	Missed Deadline Rate of Search $\left(MDR_{1}\right)$	Missed Deadline Rate of Miss $\left(MDR_{3}\right)$	Missed Deadline Rate of Tracking $\left(MDR_{2}\right)$	Missed Deadline Rate of Confirmation $\left(MDR_{4}\right)$	Utilization Rate of Time	Threat Rate of the Execution (*TRE*)
10	0.99845	0	0	0	0	0.32232	0.99796
20	0.99748	0	0	0	0.0102	0.64592	0.99797
30	0.9965	0.00218	0	0	0.03046	0.89232	0.99812
40	0.99178	0.02154	0.00165	0	0.05055	0.95784	0.99602
50	0.98601	0.0746	0	0.00073	0.06594	0.9588	0.99473
60	0.95872	0.21517	0.0016	0.0148	0.07859	0.95904	0.97849
70	0.67917	0.5866	0.09883	0.18936	0.12748	0.95896	0.80429
80	0.54817	0.67375	0.28253	0.2193	0.20579	0.95864	0.74586
90	0.48166	0.74112	0.3	0.2268	0.2119	0.95888	0.73533

**Table 4 sensors-20-00153-t004:** Performance parameters of THiPrEDF.

Parameters/the Number of Targets	*SSR*	$MDR_{1}$	$MDR_{3}$	$MDR_{2}$	$MDR_{4}$	Utilization Rate of Time	*TRE*
10	0.99845	0	0	0	0	0.32216	0.99838
20	0.99709	0	0	0	0.01107	0.6408	0.99760
30	0.99521	0	0	0	0.03209	0.88528	0.99653
40	0.99151	0.02377	0	0.00055	0.05022	0.95712	0.99588
50	0.98362	0.07450	0	0.00259	0.07353	0.9588	0.99329
60	0.95952	0.23299	0.00082	0.00945	0.08561	0.95904	0.98812
70	0.71052	0.57587	0.04827	0.15047	0.16600	0.95896	0.88677
80	0.55739	0.66999	0.23935	0.20911	0.27924	0.95864	0.78042
90	0.48949	0.73853	0.26154	0.21570	0.31462	0.95888	0.75809

**Table 5 sensors-20-00153-t005:** Performance parameters of TWD-HiPrEDF.

Parameters/the Number of Targets	*SSR*	$MDR_{1}$	$MDR_{3}$	$MDR_{2}$	$MDR_{4}$	Utilization Rate of Time	*TRE*
10	0.99846	0	0	0	0	0.32848	0.99810
20	0.99747	0	0	0	0.01056	0.64272	0.99851
30	0.99609	0	0	0	0.02919	0.89728	0.99762
40	0.99261	0.02144	0.00079	0	0.05516	0.95712	0.99672
50	0.98365	0.08347	0	0.00240	0.07066	0.9588	0.99417
60	0.96157	0.21322	0	0.01034	0.08451	0.95904	0.98746
70	0.72933	0.55824	0.08936	0.12965	0.16890	0.95896	0.89958
80	0.57517	0.68230	0.18973	0.18316	0.19851	0.95864	0.83384
90	0.49078	0.72759	0.32698	0.21438	0.29910	0.95888	0.76992

**Table 6 sensors-20-00153-t006:** The average number of targets in the tracking state for each scheduling interval under the three scheduling algorithms.

Scheduling Algorithm	Threat Area	Potential Threat Area	Nonthreat Area	Total Average
HiPrEDF	3.594	12.178	22.912	38.684
THiPrEDF	3.944	13.916	20.07	37.93
TWD-HiPrEDF	4.424	15.778	20.356	40.558

**Table 7 sensors-20-00153-t007:** The proportions of targets in different tracking areas in each scheduling interval under the three scheduling algorithms.

Scheduling Algorithm	Threat Area	Potential Threat Area	Nonthreat Area
HiPrEDF	0.093	0.315	0.592
THiPrEDF	0.104	0.367	0.529
TWD-HiPrEDF	0.109	0.389	0.502

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
