# Peer review of "A Task Scheduling Algorithm for Phased-Array Radar Based on Dynamic Three-Way Decision"

_sensors, 2019, doi:10.3390/s20010153_

Round 1
Reviewer 1 Report
Overall the manuscript is well written but can be improved.
This work presented an improved resource scheduling algorithm for arrays based on existing theory.
The methodology is rather straightforward.
Author Response
Thank you for your valuable comments.
This algorithm introduces three-way decision strategy into the task scheduling for phased radar, and it categorises targets into three threat areas, that is, threat area, non-threat area, and potential threat area according to the degree of threat of a target. It algorithm assigns different weights to different threat domains, so that a radar can reasonably allocate time during the tracking phase based on various target threat levels and threat areas. In addition, an improved adaptive threshold algorithm is proposed to obtain dynamic three-way decision to make the algorithm more flexible, adjustable according to specific environmental conditions, and therefore is more generally-applicable than other algorithms.
We have modified and improved the method and the content according to the comment.
The pseudocode of the algorithm has been added in Section 3 to help readers understand the algorithms. Further explanations of the simulation results have been added.
Reviewer 2 Report
This research is really interesting to read, and the authors have presented it well.
There are some minor mistakes you may have to fix such as replacing “Mechanical Radar” with radar with a mechanical beam-steering antenna. The algorithm in page 4 (line number 158 to 168) is not presented well. Please present it as more appropriately (Pseudocode). Figures 2, 3, 4, and 5 are not very clear. Please enlarge them. Please elaborate figure 5 more.
Author Response
Thanks for your comments.
We have carefully answered the questions one by one according to the your comments and revised the manuscript.
Please see the attachment.

Reviewer 3 Report
This paper proposes a priority assignment algorithm to improve the processing capability of phased array radars. Please consider the following concerns more clearly:
Parameters $\lambda$ and $k$ affect the results of the proposed algorithm. However, this paper does not discuss how to determine their values. In the evaluations, the parameters are set constant values. Please explain why these constant values are selected. The second level priority is called mixed criticality that has been studied for about ten years. Please read the review entitled “mixed criticality systems -- a review”. There are many novel algorithms to solve the scheduling problem for mixed criticality systems. It is helpful for the authors to improve their theoretical knowledge. The two algorithms presented in Section 3 are difficult to follow. A simple example should be given to help readers understand the algorithms. In Section 4, whether the three metrics are proposed in this paper or not? If yes, what metrics are used in related work? In Table 2, why the parameters are set the values shown in the table? If the parameters are set different values, the results will be different? From Fig. 2, 3 and 4, we know that the proposed algorithm is only slightly better than the others. If the parameters are set different values, maybe the proposed algorithm will be worse than the others. Please improve the proposed algorithm.Author Response
Thanks for your comments.
We have carefully answered the questions one by one according to the your comments and revised the manuscript.
Please see the attachment.

Round 2
Reviewer 3 Report
The added pseudocode does not help readers understand the algorithms.
1.In Line 1 of the first pseudocode, $Mini_R$ is equal to $R_x(\alpha_0,\beta_0,\gamma_0)$. However, in line 5, $R’_{x’}$ is also equal to $R_x(\alpha_0,\beta_0,\gamma_0)$. If the two values are the same, the next line always is false.
2. In Line 8, the results of $\mu(x_i)$ always are assigned to the same $\alpha$? If yes, why $\alpha$ is calculated and assigned repeatedly? $\alpha$ should be $\alpha_g$?
These are just two typical examples. In addition to these, there are many similar issues in the pseudocode. These incorrect symbols really confuse readers. Please ask for help from experienced authors.
Author Response
Thanks for your valuable comments.
We have carefully answered the questions one by one according to the your comments and revised the manuscript.
Please see the attachment.

Round 3
Reviewer 3 Report
All comments have been addressed.